# Molecular mechanism of Aurora A kinase autophosphorylation and its allosteric activation by TPX2

Adelajda Zorba[†], Vanessa Buosi[†], Steffen Kutter, Nadja Kern, Francesco Pontiggia, Young-Jin Cho[‡], Dorothee Kern*

Department of Biochemistry, Howard Hughes Medical Institute, Brandeis University, Waltham, United States

**Abstract** We elucidate the molecular mechanisms of two distinct activation strategies (autophosphorylation and TPX2-mediated activation) in human Aurora A kinase. Classic allosteric activation is in play where either activation loop phosphorylation or TPX2 binding to a conserved hydrophobic groove shifts the equilibrium far towards the active conformation. We resolve the controversy about the mechanism of autophosphorylation by demonstrating intermolecular autophosphorylation in a long-lived dimer by combining X-ray crystallography with functional assays. We then address the allosteric activation by TPX2 through activity assays and the crystal structure of a domain-swapped dimer of dephosphorylated Aurora A and TPX2[1–25]. While autophosphorylation is the key regulatory mechanism in the centrosomes in the early stages of mitosis, allosteric activation by TPX2 of dephosphorylated Aurora A could be at play in the spindle microtubules. The mechanistic insights into autophosphorylation and allosteric activation by TPX2 binding proposed here, may have implications for understanding regulation of other protein kinases.

*For correspondence: dkern@brandeis.edu

[†]These authors contributed equally to this work

Present address: [‡]New Drug Development Center, Daegu-Gyeongbuk Medical Innovation Foundation (DGMIF), Daegu, Korea

Competing interests: The authors declare that no competing interests exist.

## Introduction

The evolution of more than 500 human protein kinases from a few protein kinases in unicellular organisms allowed for the development of complexity via differential regulation (*Manning et al., 2002*). Such regulation can be achieved by autophosphorylation or interactions with other domains or binding partners. While many of the signaling cascades and their in vivo biological effectors have been well characterized, and a wealth of structural information is available (*Johnson et al., 1996*; *Huse and Kuriyan, 2002*; *Manning et al., 2002*; *Tyson et al., 2003*), the molecular mechanism whereby kinase activity is modulated is a topic of controversial debate (*Oliver et al., 2006*; *Dodson et al., 2013*). Here we investigate two fundamentally distinct regulation mechanisms by characterizing autophosphorylation of Aurora A as well as activation by TPX2 (Targeting Protein for Xklp2). Aurora A, a Ser/Thr kinase, is a key regulator of mitotic events, including mitotic entry (*Macurek et al., 2008*; *Seki et al., 2008*), centrosome maturation (*Glover et al., 1995*; *Hannak et al., 2001*; *Toji et al., 2004*; *Abe et al., 2006*; *Mori et al., 2007*), and spindle formation (*Giet et al., 2002*; *Kapitein et al., 2005*; *Tsai and Zheng, 2005*; *Koffa et al., 2006*; *Venoux et al., 2008*; *Wong et al., 2008*; *Zhang et al., 2008*). Aurora A depletion leads to cell cycle arrest, while overexpression has been found in many cancer cell lines (*Kallioniemi et al., 1994*; *Sen et al., 1997*; *Zhou et al., 1998*; *Jeng et al., 2004*). Therefore extensive interest has been recently directed towards Aurora A for anti-cancer drug development (*Aliagas-Martin et al., 2009*; *Bebbington et al., 2009*; *Cheok et al., 2010*). TPX2 recruits Aurora A to the spindle microtubules, an event that is essential in spindle formation (*Kufer et al., 2002*; *Giubettini et al., 2011*).

Autophosphorylation of T288 in the activation loop increases the catalytic activity of Aurora A (*Walter et al., 2000*; *Littlepage et al., 2002*). Intramolecular autophosphorylation has recently been suggested for Aurora A and Chk2 based on indirect kinetic measurements (*Dodson et al., 2013*)

**eLife digest** The kinase, Aurora A, is a human protein that is needed for cells to divide normally. Kinases are enzymes that control other proteins by adding phosphate groups to these proteins; however, like other kinases, Aurora A must first be activated or 'switched on' before it can do this. Aurora A kinase can be switched on in two ways: by having a phosphate group added to its 'activation loop'; or by binding to another protein called TPX2.

Also like other kinases, Aurora A can self-activate, but the details of this process are not understood. Does a single Aurora A kinase add a phosphate group to its own activation loop, or does one Aurora A kinase activate a second? Furthermore, it is not clear how binding to TPX2 can activate an Aurora A kinase without adding a phosphate group to the activation loop.

Zorba, Buosi et al. now show that Aurora A kinases that have been activated in different ways— via the addition of a phosphate group or binding to TPX2—are equally good at adding phosphate groups to other proteins. Zorba, Buosi et al. also worked out the three-dimensional shapes of the kinases activated in these two ways—since many proteins change shape when they are switched on—and found that they were also the same. Finally, it was shown that self-activation involves two Aurora A kinases binding to each other, and one kinase adding a phosphate group to the other, rather than a single kinase adding a phosphate group to itself.

Since other protein kinases can be activated in similar ways to Aurora A, the findings of Zorba, Buosi et al. might also help us to understand how other protein kinases can be switched 'on' or 'off'. And, as mutations in Aurora A have been linked to the development of cancer, uncovering how this kinase is controlled could help efforts to design new drugs to treat this disease.

adding to the controversy by disagreeing with the intermolecular mechanism proposed for other protein kinases (*Oliver et al., 2006*, *2007*; *Pike et al., 2008*; *Lee et al., 2009*).

A second puzzling result has also been reported recently. It was shown that in vivo during mitosis, TPX2-bound Aurora A at the spindle microtubules is dephosphorylated at the crucial T288 (*Toya et al., 2011*). Since T288-dephosphorylated Aurora A exhibits very low kinase activity, a second kinase-independent function of Aurora A was postulated (Littlepage, 2002). There is evidence suggesting that TPX2 also plays an active role in upregulating Aurora A activity, however the interplay between the two distinct activation mechanisms, phosphorylation and TPX2-binding, is not well understood (*Kufer et al., 2002*; *Carmena and Earnshaw, 2003*; *Eyers and Maller, 2003*; *Eyers et al., 2003*; *Kufer et al., 2003*; *Trieselmann et al., 2003*; *Tsai et al., 2003*; *Bayliss et al., 2004*; *Brunet et al., 2004*; *Eyers and Maller, 2004*; *Ozlu et al., 2005*; *Tsai and Zheng, 2005*; *Anderson et al., 2007*). Here we address both controversies by directly measuring autophosporylation and by characterizing the molecular mechanism of Aurora A regulation by TPX2.

## Results and discussion

### Phosphorylation of T288 in Aurora A or TPX2 binding results in comparable increases in catalytic activity

While it is generally accepted that phosphorylation of a Ser/Thr in the activation loop activates Ser/Thr kinases, this regulation has not been characterized quantitatively. Part of the difficulty consists in obtaining a dephosphorylated protein, since *Escherichia coli*-produced kinases are heavily phosphorylated due to autophosphorylation and phosphorylation by *E. coli* kinases during expression (*Enami and Ishihama, 1984*) (*Figure 1A*).

Aurora A was co-expressed with the generic Ser/Thr/Tyr phosphatase lambda (λPP) and treated again with λPP after purification to ensure complete dephosphorylation (*Figure 1A*). We then measured the enzymatic activity of phosphorylated versus dephosphorylated Aurora A towards AP, a peptide encompassing residues 281–293 of the kinase's activation segment. The experiments were conducted under saturating conditions of peptide, ATP, and Mg$^{2+}$, and we ensured that the singly and heavily phosphorylated kinases exhibited similar kinetics (*Figure 1—figure supplement 1 and 2*).

Phosphorylated Aurora A catalyzes AP phosphorylation 100-fold faster than the dephosphorylated kinase (1.0 ± 0.2 s$^{-1}$ vs 0.01 ± 0.005 s$^{-1}$; *Figure 1B*). Since dephosphorylated Aurora A can also

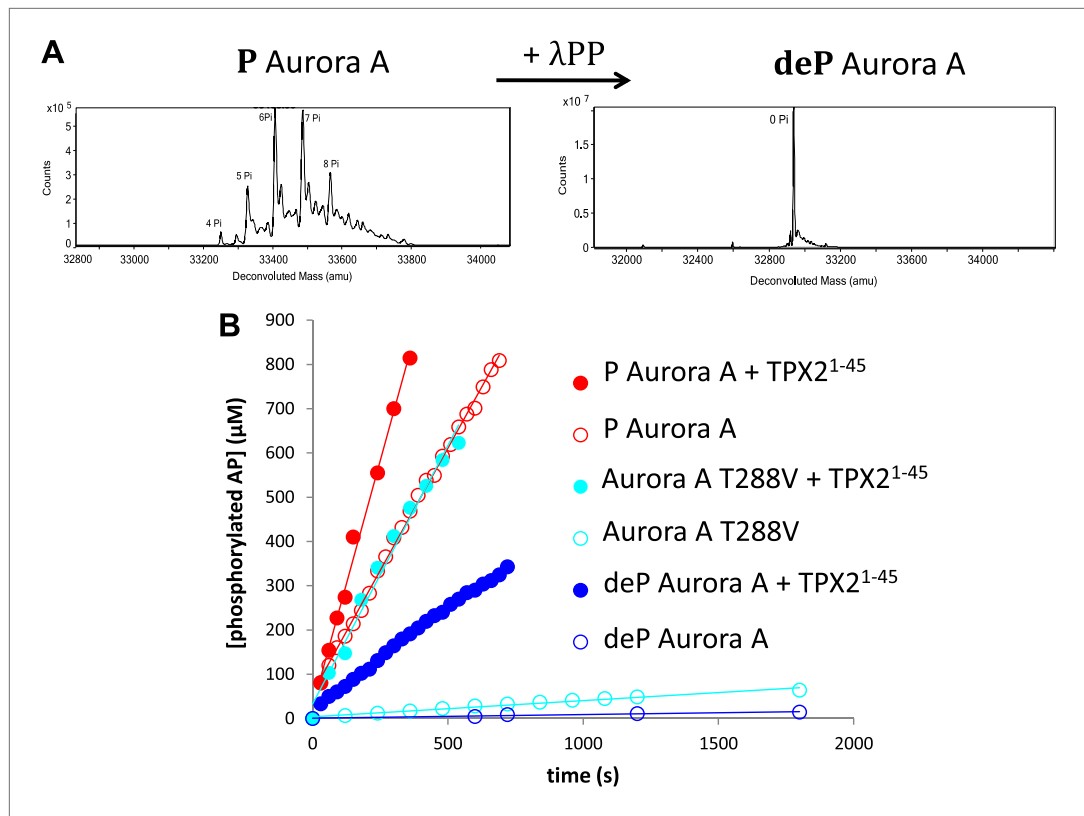

**Figure 1**. TPX2[1–45] drastically accelerates the kinetics of the dephosphorylated form of Aurora A kinase. (**A**) Mass spectrometry data of heavily phosphorylated (P) and dephosphorylated (deP) Aurora A. The dephosphorylated protein was obtained after treatment of heavily phosphorylated, *Escherichia coli*-produced Aurora A with λ-protein phosphatase (λPP). (**B**) AP phosphorylation by dephosphorylated ($\bigcirc$, 0.01 ± 0.005 s$^{-1}$) or T288V mutant Aurora A ($\bigcirc$, 0.05 ± 0.002 s$^{-1}$) is increased by up to 50-fold ($\bullet$, 0.5 ± 0.1 s$^{-1}$) and 25-fold ($\bullet$, 1.2 ± 0.1 s$^{-1}$), respectively, in the presence of TPX2[1–45]. This rate is comparable to the kinetics of phosphorylated Aurora A in the absence of TPX2[1–45] ($\bigcirc$, 1.0 ± 0.2 s$^{-1}$). Phosphorylated Aurora A shows up to a twofold increase in AP kinetics in the presence of TPX2[1–45] ($\bullet$, 2.3 ± 0.2 s$^{-1}$). Reactions are carried out in the presence of 1 μM protein, 50 μM TPX2[1–45], 5 mM ATP, and 1 mM AP in assay buffer (20 mM TrisHCl, 200 mM NaCl, 20 mM MgCl$_2$, 3% (vol/vol) glycerol, 1 mM TCEP, pH 7.50) at 25°C. Phosphorylated peptide production was monitored by reverse phase-high performance liquid chromatography (RP-HPLC).

The following figure supplements are available for figure 1:

**Figure supplement 1**. Kinase assays were conducted under saturating conditions of peptide and ATP.

**Figure supplement 2**. Aurora A exhibits the same activity towards AP whether the protein is phosphorylated on multiple sites or singly phosphorylated on T288.

**Figure supplement 3**. Aurora A kinase autophosphorylation and substrate phosphorylation were simultaneously followed either in the absence or presence of TPX2[1–45].

**Figure supplement 4**. TPX2[1–45] drastically accelerates the kinetics of the dephosphorylated-like Aurora A species irrespective of the nature of the peptide used.

**Figure supplement 5**. Representative RP-HPLC time traces during AP phosphorylation.

**Figure supplement 6**. Dose-dependence of the concentration of TPX2[1–147] on the phosphorylation kinetics of AP by Aurora A T288V.

autophosphorylate during this experiment, we not only quantified autophosphorylation during the reaction time frame (*Figure 1–figure supplement 3*), but we also designed a second experiment that eliminates this competing reaction by mutating T288 to V. This mutant shows comparable low activity to the dephosphorylated kinase ($k_{cat}^{AP}$ = 0.05 ± 0.002 s$^{-1}$) serving as a control for quantifying the rate acceleration provided by T288 phosphorylation.

Does TPX2 activate Aurora A to the same extent as phosphorylation or are both activation mechanisms additive? We used TPX2$^{1–45}$ in our studies since this fragment was shown to be sufficient for kinase activation (*Bayliss et al., 2003*; *Brunet et al., 2004*). Our data suggest the first scenario since AP phosphorylation is stimulated by 50- and 25-fold by TPX2$^{1–45}$ for the dephosphorylated Aurora A and T288V Aurora A, respectively (*Figure 1B*). The rates of dephosphorylated Aurora A plus TPX2$^{1–45}$ are comparable to that of phosphorylated Aurora A alone. Addition of TPX2 to phosphorylated Aurora A results in only a twofold increase in $k_{cat}$. This effect is independent of the nature of the peptide used (*Figure 1–figure supplement 4*).

## Dephosphorylated Aurora A kinase is in the active conformation when bound to TPX2$^{1–45}$

Having shown kinetically that TPX2 is sufficient to activate dephosphorylated Aurora A similarly to T288 phosphorylation, we next studied the underlying molecular mechanism. We first solved the crystal structure of dephosphorylated Aurora A in the absence and presence of TPX2$^{1–45}$ and bound to an ATP-mimic (β,γ-methyleneadenosine 5' triphosphate, AMPPCP). The AMPPCP-bound, dephosphorylated Aurora A is monomeric and in an inactive conformation, similar to previously solved structures of the same protein bound to adenosine (PDB ID 1MUO [*Cheetham et al., 2002*]) or AMPPNP (PDB ID 2C6D [*Heron et al., 2006*]). In contrast, the TPX2-bound Aurora A structure reveals a dimer made by two molecules of Aurora A, TPX2, and AMPPCP in the asymmetric unit (*Figure 2A*, *Table 1*).

Since the classic bilobal fold of protein kinases is by now well known from many elegant structural studies (*Cheetham, 2002*; *Bayliss et al., 2003*), and our structure in the absence of TPX2 does not provide new information, we will only discuss novel insights gained from the TPX2-bound dephosphorylated Aurora A dimer.

Superposition of the Aurora A monomers shows that they are similar, but not identical within the heterodimer, with implications for autophosphorylation discussed below (*Figure 2B*). Interestingly, both monomers are in an active conformation despite being dephosphorylated (*Figure 2C*), a feature that has never been seen before in Aurora kinases. Comparison of dephosphorylated Aurora A in the absence or presence of TPX2 reveals subtle but significant interactions by which TPX2 stabilizes the active form of the kinase (*Figure 2C*).

First, binding of TPX2 causes a slight rotation of the αC-helix towards the catalytic center, thus allowing for the conserved, stabilizing E181-K162 ion pair to form. The αC-helix rotation also results in movement of F275 of the conserved DFG motif from the DFG-out into the DFG-in position (*Martin et al., 2012*) (*Figure 2C*, bottom inset). This positions the catalytic D274 in the correct orientation to carry out phosphoryl transfer.

Second, movement of F275 initiates a cascade of side chain interactions that result in the completion of the regulatory spine originating from the αF-helix (R-spine; *Figure 2C*, bottom inset). Identification of the completed R-spine, a hallmark of an active kinase, is based on the Local Spatial Patterns (LSP) alignment, a bioinformatics tool developed by the Taylor laboratory (*Kornev et al., 2008*). From these structural features it appears that Aurora A, despite being dephosphorylated, is in an active conformation when bound to TPX2. We note that *Bayliss et al. (2003)* propose that the structure of phosphorylated Aurora A represents a partially active state because the authors interpret the activation segment to be in an inactive conformation as defined by the exposure of pThr288 to the solvent (*Bayliss et al., 2003*). We would rather interpret that structure as an active state based on all hallmarks for active kinases (*Kornev et al., 2008*), in agreement with our activity data (*Figure 1B*).

## Does active, domain-swapped Aurora A dimer capture an enzyme/substrate complex for autophosphorylation?

The final signatures for an active state Ser/Thr kinase involve changes around the activation segment. Formation of the β6/β9 antiparallel β sheets, that are not present in the inactive kinase, prime the MgATP for catalysis together with the other conformational changes described above. Finally, the correct positioning of D256 for activating the hydroxyl of the substrate in active Aurora A is achieved

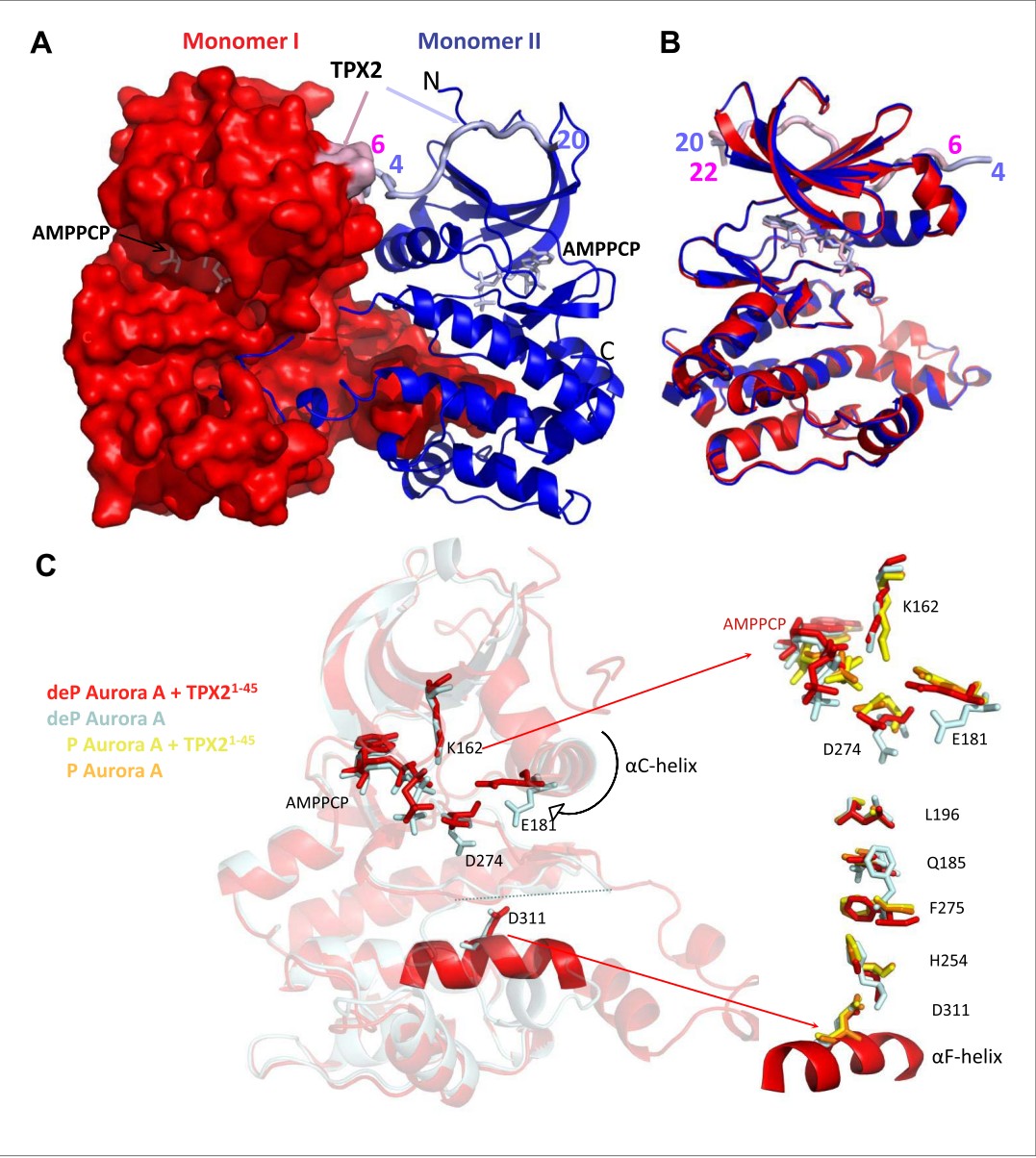

**Figure 2**. Dephosphorylated Aurora A adopts an active conformation in the presence of TPX2[1–45]. (**A**) Dephosphorylated Aurora A + TPX2[1–45] + AMPPCP and (**B**) superposition of Aurora A moieties. (**C**) A detailed view of structural elements that define an active Aurora A kinase: the nucleotide binding region (top inset) and the regulatory spine (bottom inset). Dephosphorylated (deP) Aurora A in the presence of TPX2 (red, PDB ID 4C3P) superposes very well to the phosphorylated (P) Aurora A either in the absence (orange, PDB ID 1OL7) or presence of TPX2 (yellow, PDB ID 1OL5). For comparison, dephosphorylated Aurora A alone (light blue, PDB ID 4C3R) shows the characteristic features of an inactive kinase.

by a hydrogen bond network to T292, which in turn H-bonds to K258 with distances shown in *Figure 3A*. All these features are fully conserved among eukaryotic Ser/Thr kinases (*Nolen and Taylor, 2004*).

Curiously, the last two hallmarks of the active state are created by a swap of the activation segments of Aurora A resulting in a dimer conformation that has not been previously reported for this protein. In the C-lobe, W313 that would typically interact with P297 and P298 in monomeric Aurora A, now interacts with the identical Pro residues of the other Aurora A monomer. Lastly, the E299–R371 salt bridge is also domain swapped (*Figure 3A*).

Since autophosphorylation is considered a crucial regulation mechanism for protein kinases, the biological meaning of such a dimer structure becomes immediately apparent. While trans-activation through

**Table 1.** Data collection and refinement statistics

| | deP Aurora A + AMPPCP | deP Aurora A + AMPPCP + TPX2 |
|---|---|---|
| Data collection | | |
| Space group | P 6₁ 2 2 | P 2₁ 2₁ 2₁ |
| Cell dimensions | | |
| $a, b, c$ (Å) | 83.47, 83.47, 172.63 | 49.93, 86.72, 153.55 |
| α, β, γ (°) | 90, 90, 120 | 90, 90, 90 |
| Resolution (Å) | 86.3–2.79 (2.87–2.79) | 86.7–2.69 (2.76–2.69) |
| $R_{merge}$ | 0.08 (2.08) | 0.26 (3.51) |
| $I/\sigma$ | 19.8 (1.8) | 8.0 (2.3) |
| Completeness (%) | 100 (100) | 100 (100) |
| Redundancy | 15.7 (16.7) | 6.9 (7.2) |
| Refinement | | |
| Resolution (Å) | 55.4–2.79 (2.87–2.79) | 47.5–2.69 (2.76–2.69) |
| No. reflections | 8459 | 18104 |
| $R_{work}/R_{free}$ | 0.221/0.306 (0.327/0.451) | 0.201/0.289 (0.284/0.400) |
| No. atoms | | |
| Protein | 2074 | 4574 |
| Ligand/ion | 31 | 67 |
| Water | 0 | 21 |
| B-factors | | |
| Protein | 100.3 | 54.6 |
| Ligand/ion | 109.1 | 62.5 |
| Water | NA | 43.3 |
| R.m.s deviations | | |
| Bond lengths (Å) | 0.010 | 0.011 |
| Bond angles (°) | 1.54 | 1.53 |
| PDB ID | 4C3R | 4C3P |

Values in parentheses correspond to the highest-resolution shell.
deP: dephosphorylated; PBD, Protein Data Bank.

domain-swapped dimers seen in X-ray structures has been proposed for a number of kinases (*Figure 3–figure supplement 1*) (*Oliver et al., 2006*, *2007*; *Pike et al., 2008*; *Lee et al., 2009*), a recent report on Aurora A and Chk2 suggesting a strict intramolecular autophosphorylation of the activation segment for both enzymes has fueled controversy on this topic (*Dodson et al., 2013*). Another puzzle is the fact that the activation-segment swapped dimers in the literature (*Figure 3–figure supplement 1*) show both monomers to be either in the inactive or active kinase conformations, and a number of these structures are either bound to an inhibitor or already phosphorylated.

In contrast we find an asymmetric dimer within the unit cell with monomer I showing complete electron density for the activation segment while monomer II is missing residues 283–288 (*Figure 2B*, red and blue, respectively). In addition, only monomer I has the perfect geometry of the conserved hydrogen bond network found in a catalytically prone kinase between residues D256, K258, and T292 (*Figure 3A*). Therefore we speculated that monomer I may be the enzyme molecule recognizing monomer II as its substrate.

As a first (and only crude) test of this hypothesis, we used targeted MD (TMD) simulations to model the target hydroxyl for autophosphorylation of the domain-swapped activation segment towards the γ-phosphate of AMPPCP (*Figure 3B*) to address the question whether it is even physically possible that the hydroxyl can approach the γ-phosphate of AMPPCP. Only T288 of monomer II (proposed as the substrate molecule above) with a more flexible activation loop segment can be rearranged at a close distance to the γ-phosphate of AMPPCP without displacing the AMPPCP from its original position (*Figure 3B*).

## Functional evidence for intermolecular autophosphorylation within the swapped dimer

Neither the crystallographic dimer nor the TMD simulations are compelling evidence for intermolecular autophosphorylation within such a swapped dimer. To answer the first obvious question whether a swapped dimer exists in solution, we performed small-angle X-ray scattering (SAXS) and sedimentation velocity analytical ultracentrifugation (AUC) experiments on dephosphorylated Aurora A in the absence and presence of TPX2 (*Figure 4*). A significant amount of dimer was detected for Aurora A and the relative concentration was independent of the presence of TPX2. Importantly, the fact that separate peaks for the dimer and monomer were observed in a sedimentation velocity run reveals a slow interconversion rate between dimer and monomer. Fitting of the SAXS data required inclusion of 9% dimer of the shape of our X-ray structure, supporting the notion that the swapped dimer seen in X-ray crystallography exists in solution in agreement with the AUC data. Protein solubility prohibited determination of the $K_D$, but from both the concentration dependence observed in AUC as well as SAXS experiments one can estimate that the $K_D$ is above 300 μM.

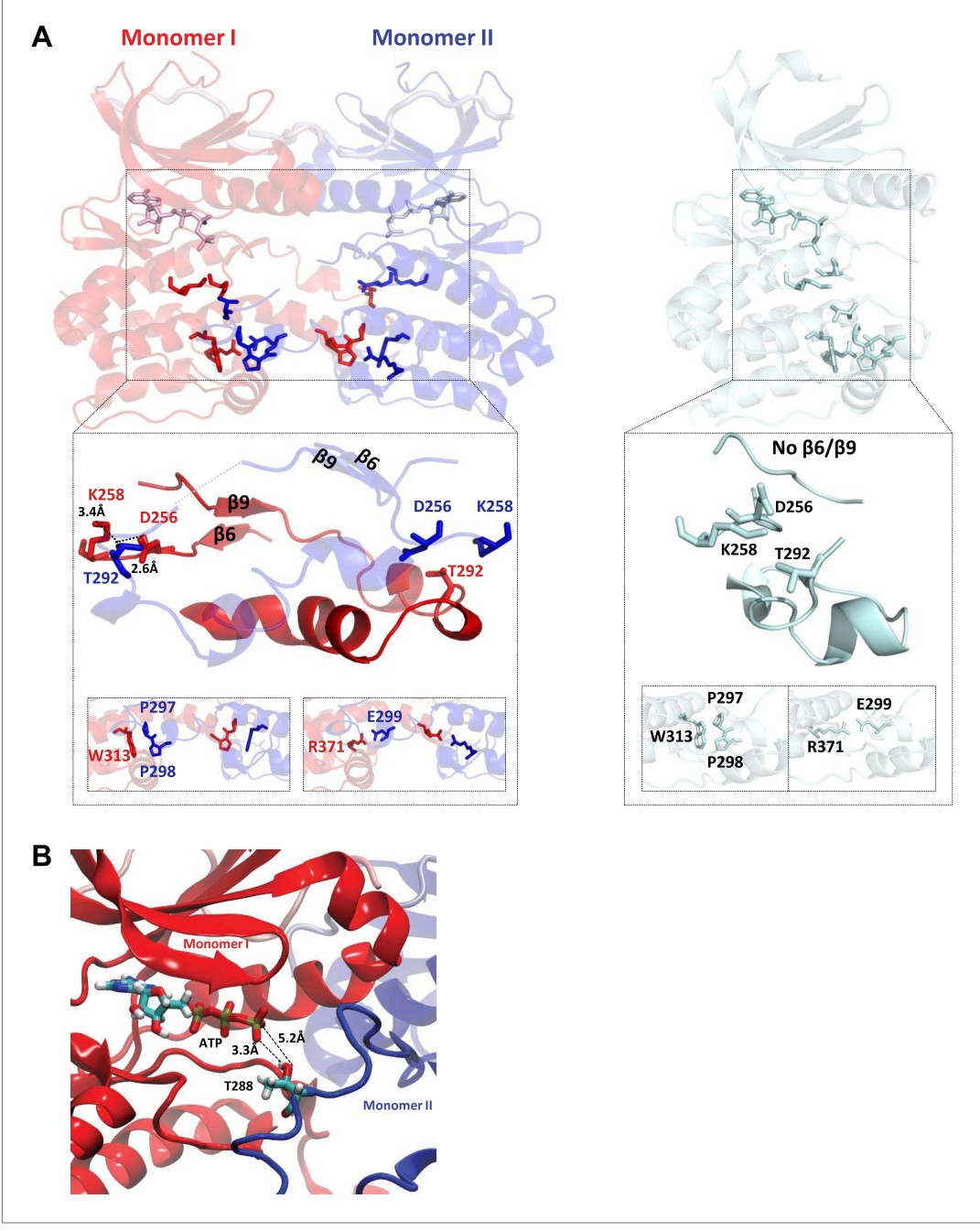

**Figure 3**. TPX2-bound domain-swapped Aurora A captures an enzyme–substrate complex. (**A**) Left: in the presence of TPX2$^{1–45}$, the N-terminal (β6 and β9 sheets) anchor point of the activation loop is present in both monomers whereas the C-terminal H-bond contacts typical for a fully active kinase (between D256/K258 and T292-OH) are only visible for the enzyme monomer in red, which we therefore define as the enzyme molecule. Right: for comparison, in the absence of TPX2$^{1–45}$, the N- and C-terminal anchor points are not present and the protein is in an inactive state. Interactions that further stabilize the swapped dimer (W313-P297/P298 and R371-E299) are shown in the bottom inset, highlighting that these intermolecular interactions (left) are identical to the corresponding intramolecular interactions (right). (**B**) The loop spanning residues 283–288 in monomer II, for which there was too weak electron density, was remodeled using the software Modeller and biased molecular dynamics. The loop can be arranged by TMD so that the distance between T288 of monomer II and γ-phosphate of AMPPCP of monomer I is compatible with phosphoryl transfer.

The following figure supplements are available for figure 3:

**Figure supplement 1**. Comparison of the dimeric Aurora A + TPX2 structure with other domain-swapped Ser/Thr kinases.

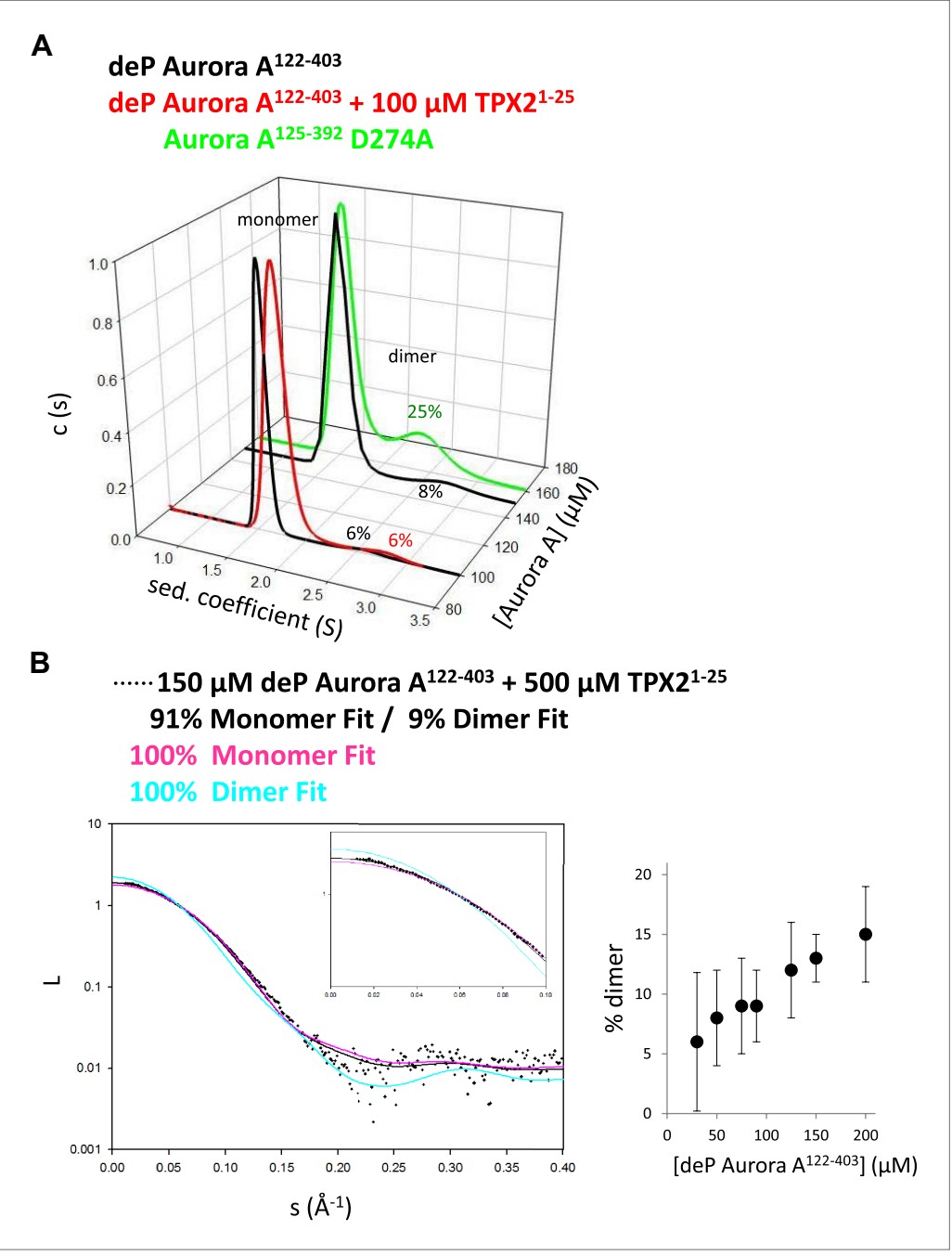

**Figure 4**. TPX2-bound domain-swapped Aurora A forms a stable dimer in solution. (**A**) Sedimentation velocity analytical ultracentrifugation data show discrete peaks for monomer and dimer. TPX2[1–25] does not increase the percentage of Aurora A dimer in solution. It is unclear why there is an increased dimer concentration for the kinase-dead Aurora A D274A mutant. (**B**) Small-angle X-ray scattering (SAXS) data show an increase in dimer concentration with increased Aurora A amounts. All data were collected in the presence of 500 μM AMPPCP in kinase assay buffer. deP, dephosphorylated.

The existence of the dimer in solution is essential but not sufficient for an intermolecular autophosphorylation mechanism. To directly investigate the mechanism, functional assays were performed. First, we designed two Aurora A constructs that differed in activity and length. D274A is a kinase-impaired version (**Wan et al., 2008**) that is unable to autophosphorylate within our reaction time frame (**Figure 5A**, middle panel). Since the N- and C-terminal truncations do not affect activity

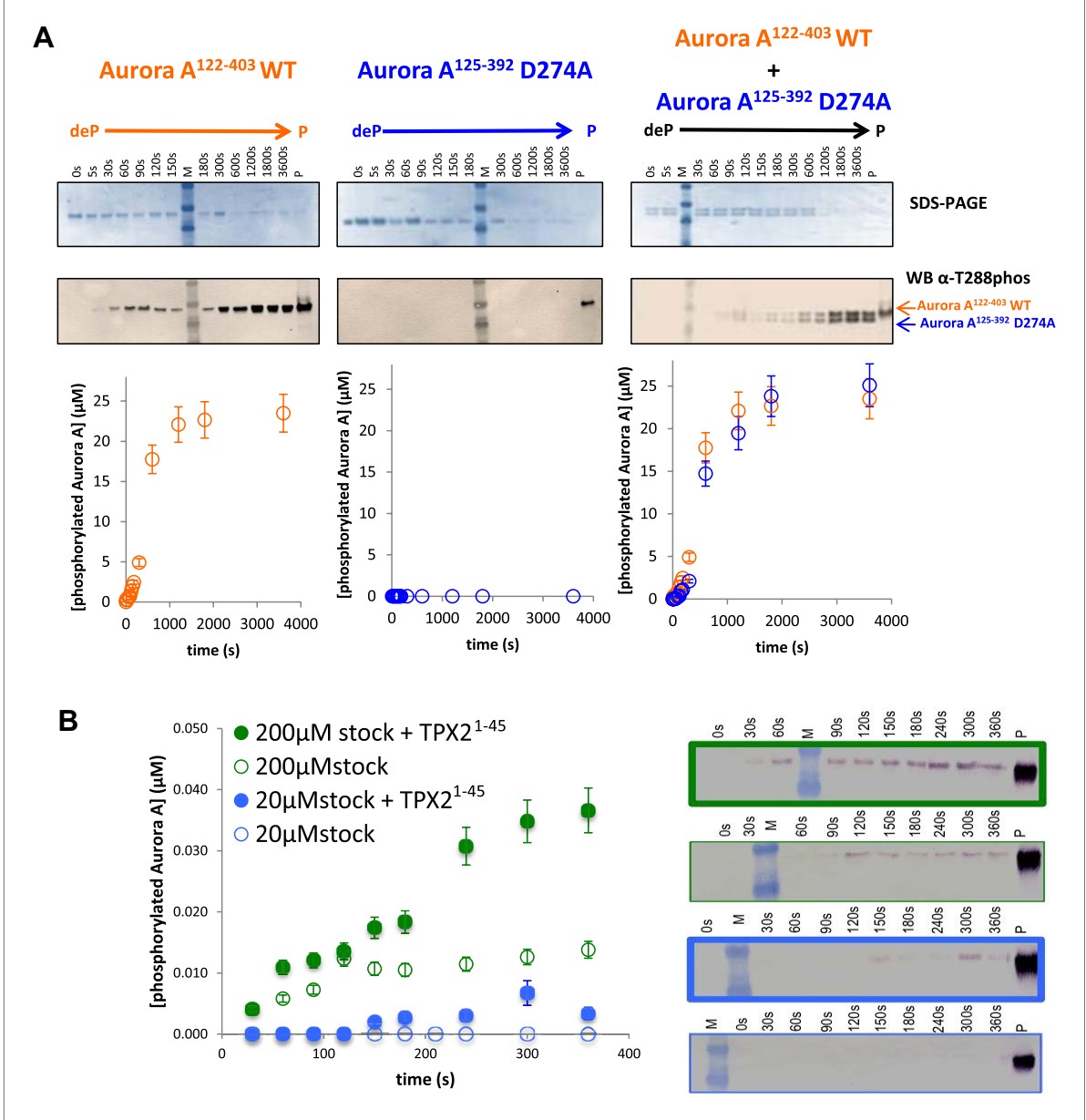

**Figure 5.** Mechanism of autophosphorylation. (**A**) The kinetics of autophosphorylation was monitored by SDS-PAGE and Western blot of 25 µM Aurora A[122–403] WT or 25 µM Aurora A[125–392] D274A. WT Aurora A can phosphorylate catalytically dead D274A Aurora A intermolecularly. To account for Aurora A's dynamic range, time points up to 300 s were diluted 50-fold and the rest of the time points were diluted 225-fold. (**B**) Dilution to 1 µM protein from a stock solution of 200 µM Aurora A ± TPX2 shows much faster autophosphorylation kinetics than from a lower concentrated stock solution (20 µM ± TPX2) revealing that autophosphorylation occurs within the long-lived dimer. All experiments were carried out at 25°C in kinase assay buffer in the presence of 5 mM ATP.

The following figure supplements are available for figure 5:

**Figure supplement 1.** Dephosphorylated Aurora A[125–392] can autophosphorylate as efficiently as Aurora A[122–403].

(*Figure 5–figure supplement 1*), this trick allowed for easy simultaneous detection of wild type and kinase-impaired Aurora A. While Aurora A[125–392] D274A cannot phosphorylate itself, equal concentrations of dephosphorylated wild type Aurora A[122–403] and Aurora A[125–392] D274A lead to comparable phosphorylation rates for both proteins, demonstrating an intermolecular autophosphorylation mechanism for Aurora A (*Figure 5A*, right panel).

The fact that the dimer is long lived as identified by AUC triggered a second independent test of the functional relevance of this swapped dimer. The rate of autophosphorylation was measured for samples of 1 μM Aurora A prepared by dilution from 200 μM and 20 μM stock solutions. Although the final protein concentration in both samples is the same, much faster autophosphorylation was detected for the sample diluted from the more highly concentrated stock solution (*Figure 5A*). Since the latter sample contains a higher concentration of this slowly dissociating dimer, this experiment directly demonstrates dimer-dependent autophosphorylation. TPX2 strongly accelerates autophosphorylation, in agreement with previous reports (*Eyers et al., 2003*), and as seen for peptide phosphorylation (*Figure 1B* and *Figure 1—figure supplement 4*). Notably, this intermolecular autophosphorylation again proceeds via the long-lived swapped dimer (*Figure 5B*). We note that although necessary, the dimer is not sufficient for autophosphorylation, because the catalytically impaired D274A mutant shows a high percentage of dimer (*Figure 4A*). Clearly, TPX2 triggers a conformational change that results in a catalytically active dimer.

As the third line of evidence, we aimed at designing a mutation that would weaken the swapped dimer formation and therefore autophosphorylation without compromising the kinase activity of the phosphorylated monomer towards peptides. Realization of this thought experiment is challenging because most of the intermolecular interactions for the dimer are present as corresponding intramolecular contacts in the monomer (*Figure 3A*). We rationalized that a C290A mutation could work because C290 of one monomer contacts Y334 of the αG-helix of the other monomer, while in phosphorylated monomeric Aurora A, C290 contacts K143, W277, L289, and G291 (*Figure 6A,B*). The C290A mutation indeed results in a primarily monomeric form (*Figure 6C*) that also has severely impaired autophosphorylation activity (*Figure 6D*). However, once phosphorylated, C290A has nearly normal catalytic activity towards the AP substrate (*Figure 6E*), buttressing the functional role of the swapped WT dimer for autophosphorylation. Particularly striking is the observation that in a 1:1 mixture of dephosphorylated WT and C290A mutant, WT autophosphorylation precedes C290A phosphorylation in the early reaction time course (*Figure 6D*, right and inset) but phosphorylation kinetics are identical at later time points. Such kinetic behavior is expected for a model of initial autophosphorylation between two dephosphorylated molecules within the swapped dimer and the subsequent taking over of intermolecular autophosphorylation by newly phosphorylated enzyme molecules. This latter reaction is much faster, suggesting atomistic differences in comparison to the swapped dimer reaction. The C290A mutant is incapable of forming a hybrid swapped dimer between one molecule each of dephosphorylated C290A and WT protein, explaining the lag in its phosphorylation kinetics relative to WT.

## TPX2[1–25] is necessary and sufficient for binding to Aurora A

TPX2[1–43] had been identified as essential for Aurora A activation and protection from PP1, PP2A, or λPP-directed dephosphorylation (*Eyers et al., 2003*; *Tsai et al., 2003*; *Satinover et al., 2004*). This finding was further substantiated by an X-ray structure of TPX2[1–43] bound to phosphorylated Aurora A (*Bayliss et al., 2003*) (*Figure 7—figure supplement 1*). In this structure, TPX2[6–23] was seen in an extended conformation, whereas TPX2[30–43] formed a regular helix that was proposed to be crucial in protecting Aurora A from phosphatase-mediated T288 dephosphorylation. Surprisingly, in our dimer structure, we could only visualize TPX2[4/6–20/22] (*Figure 2A,B* and *Figure 7—figure supplement 1*), and the previously seen helical part was completely missing.

To investigate this unexpected result, we designed and functionally characterized the interplay between two peptides, TPX2[1–25] and TPX2[25–45], and Aurora A. First, isothermal titration calorimetry (ITC) showed that TPX2[1–25] bound to Aurora A with the same affinity as longer versions (TPX2[1–147] or TPX2[1–45]) and did not discriminate between the phosphorylated and dephosphorylated states of the protein (*Figure 7A*). On the other hand, no signal was detected for TPX2[25–45] with Aurora A in ITC. Second, TPX2[1–25] binding could trigger an increase in activity of dephosphorylated Aurora A towards peptides and autophosphorylation (data not shown). In contrast, TPX2[25–45] had no effect on Aurora A activity. Third, TPX2[1–25] could protect Aurora A from λPP-directed dephosphorylation as well as TPX2[1–45], whereas no protection was found for TPX2[25–45] (*Figure 7B*). These functional data suggest that the first 25 amino acids of TPX2 are primarily responsible for both activation of the enzyme and protection from dephosphorylation.

The functional role for the first 25 residues in TPX2 makes sense from a comparison to the regulation of other human protein kinases (*Gold et al., 2006*; *Kannan et al., 2007*; *Keshwani et al., 2012*;

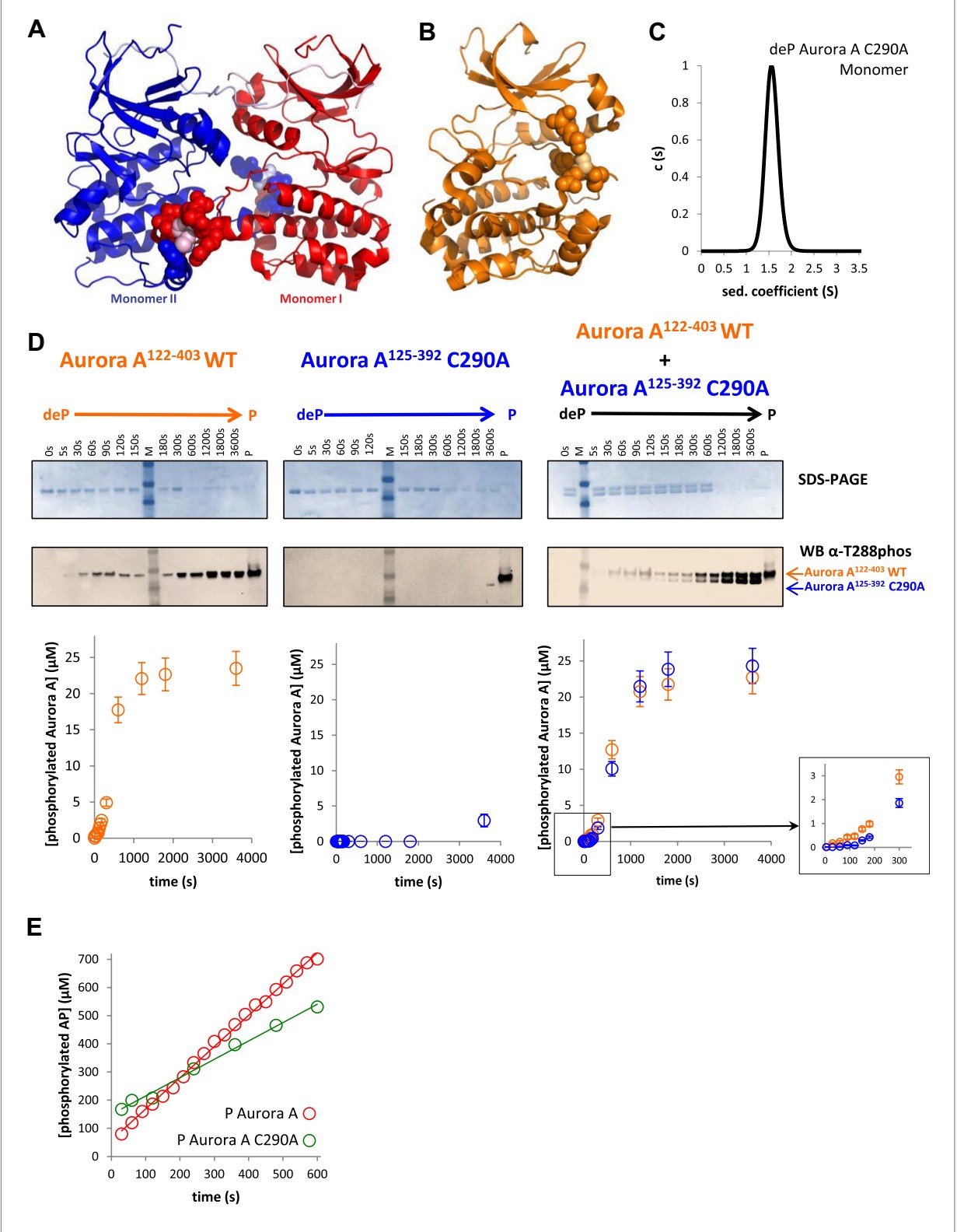

**Figure 6**. A mutant at the dimer interface (C290A) disrupts the swapped-dimer formation and autophosphorylation without affecting the activity of the phosphorylated Aurora A C290A monomer. (**A**) C290 of monomer I (light pink spheres) packs against Y334 of the αG-helix of monomer II. Other residues within a 4.5 Å radius of C290 in monomer I are shown as red spheres. (**B**) In the monomeric, phosphorylated Aurora A (PDB ID 1OL7), C290 (light orange)

*Figure 6. Continued on next page*

*Figure 6. Continued*

does not contact the αG-helix (contact residues within a 4.5 Å radius are shown as orange spheres). (**C**) Sedimentation velocity analytical ultracentrifugation of 100 μM dephosphorylated (deP) Aurora A C290A + 500 μM AMPPCP in kinase assay buffer shows that this protein is predominately monomeric in solution. (**D**) The kinetics of autophosphorylation was monitored by SDS-PAGE and Western blot of 25 μM Aurora A$^{122–403}$ WT or 25 μM Aurora A$^{125–392}$ C290A as described in *Figure 5*. C290A mutant has impaired autophosphorylation, but is readily phosphorylated by WT Aurora A. (**E**) Activity of the phosphorylated (P), monomeric C290A towards AP peptide (0.7 ± 0.1 s$^{-1}$) is comparable to that of the WT protein (1.0 ± 0.2 s$^{-1}$).

*Arencibia et al., 2013*). This specific interaction between a hydrophobic groove at the junction of the αB/αC-helices in the N-lobe of kinases and a short sequence called 'the hydrophobic motif' (Tyr/Phe-X(X)-Tyr/Phe) from either another partner or from its own C-terminal tail, seems to be a conserved regulatory mechanism across the AGC family (*Figure 7–figure supplement 2*) (*Zhang et al., 1994*). For CMGC kinases, the hydrophobic groove is occupied by activating cyclin proteins, as originally reported for cyclin binding to the N-lobe of Cdk2 which leads to reorientation of the activation loop and αC-helix into an active conformation (*Jeffrey et al., 1995*) similar to the effect of TPX2 on Aurora A. In MAPK, activity is increased by binding of the proteins' own C-terminal tails into this hydrophobic groove (*Baumli et al., 2008*). In the tyrosine kinase (TK) family, an N-terminal fragment that precedes the kinase domain of c-KIT, MET, and Ephrin nestles inside the hydrophobic pocket, and autoinhibits the kinases (*Chan et al., 2003*; *Mol et al., 2003*, *2004*; *Davis et al., 2008*; *Eathiraj et al., 2011*). In the EGFR members of the TK family, this hydrophobic motif is used as a docking point for kinase activation through dimerization (*Zhang et al., 2006*; *Jura et al., 2011*). This striking conservation of a very specific recognition mechanism across evolutionarily divergent kinase families suggests that Y8 and Y10 of TPX2 nestled inside the hydrophobic αB/αC pocket in Aurora A are likely the key triggers for kinase activation (*Figure 7–figure supplement 2*).

## Conclusions

In this work, we characterized two distinct molecular activation mechanisms of Aurora A: autophosphorylation and allosteric activation through TPX2 binding. Because of the controversy about the autophosphorylation mechanism of protein kinases (*Oliver et al., 2006*, *2007*; *Pike et al., 2008*; *Lee et al., 2009*; *Lochhead, 2009*; *Dodson et al., 2013*; *Hu et al., 2013*), we felt the need for multiple lines of evidence. Our aim was to determine the structure of a dephosphorylated enzyme/substrate complex 'ready for autophosphorylation'. The swapped dimer is indeed asymmetric, with one monomer playing the role of the enzyme and the other that of the substrate. In the substrate molecule, the hydroxyl group of T288 is in principle capable of reaching the γ-phosphate of AMPPCP bound to the enzyme monomer as shown by TMD simulations. Domain-swapped dimers have been solved for a number of protein kinases and questioned for their relevance or crystal artifacts (*Dodson et al., 2013*). To address such a critique directly, we showed a long-lived dimer in solution. Importantly, using three biochemical tricks of (i) a mixture between wild type and dead mutant and (ii) a serial dilution, and (iii) a C290A mutant with severely impaired dimer formation and autophosporylation but nearly normal peptide phosphorylation activity in its phosphorylated monomeric state, we could directly demonstrate an intermolecular autophosphorylation mechanism within this long-lived dimer.

Our combined biochemical, thermodynamic, structural, and computational data resolve the controversy about the molecular mechanism(s) of autophosphorylation in Aurora A by directly *measuring the process of autophosphorylation* and linking it to a long-lived functional dimer of dephosphorylated Aurora A. This is in sharp contrast to the mechanism of intramolecular autophosphorylation put forward by *Dodson et al. (2013)* based on *measuring the kinetics of peptide phosphorylation* using an automated microchip assay as indirect readout, and not by monitoring autophosphorylation. While the experiments in *Dodson et al. (2013)* are conceptually correct, there are a number of errors in the experimental setup and data analysis. Michaelis–Menten equations applied to analyze all data cannot be used because Michaelis–Menten conditions are not met, since the peptide concentration of 3 μM is not much higher than the enzyme concentrations used. Competition of free enzyme for the peptide versus another enzyme molecule is not included in the scheme. The peptide concentration is far below its $K_M$ and ATP hydrolysis is faster under these conditions than peptide phosphorylation producing ADP concentrations that cause significant and time-dependent enzyme inhibition (DK, unpublished). Finally, knowledge of the slow dissociation of the dimer and its consequences for the measured kinetics (*Figure 5B*) exposes another source for incorrect data interpretation since the experiments

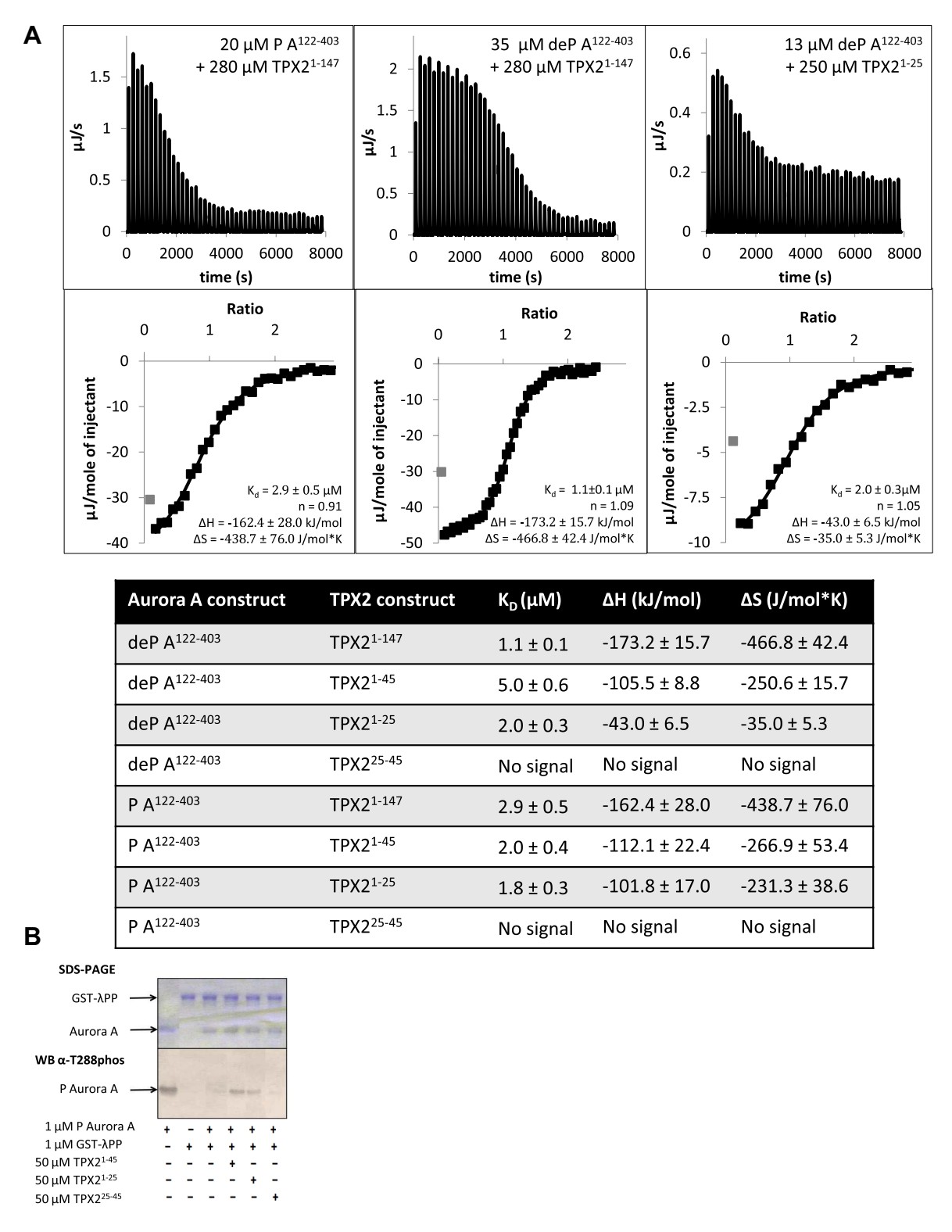

| Aurora A construct | TPX2 construct | $K_D$ (µM) | ΔH (kJ/mol) | ΔS (J/mol*K) |
|---|---|---|---|---|
| deP A$^{122-403}$ | TPX2$^{1-147}$ | 1.1 ± 0.1 | -173.2 ± 15.7 | -466.8 ± 42.4 |
| deP A$^{122-403}$ | TPX2$^{1-45}$ | 5.0 ± 0.6 | -105.5 ± 8.8 | -250.6 ± 15.7 |
| deP A$^{122-403}$ | TPX2$^{1-25}$ | 2.0 ± 0.3 | -43.0 ± 6.5 | -35.0 ± 5.3 |
| deP A$^{122-403}$ | TPX2$^{25-45}$ | No signal | No signal | No signal |
| P A$^{122-403}$ | TPX2$^{1-147}$ | 2.9 ± 0.5 | -162.4 ± 28.0 | -438.7 ± 76.0 |
| P A$^{122-403}$ | TPX2$^{1-45}$ | 2.0 ± 0.4 | -112.1 ± 22.4 | -266.9 ± 53.4 |
| P A$^{122-403}$ | TPX2$^{1-25}$ | 1.8 ± 0.3 | -101.8 ± 17.0 | -231.3 ± 38.6 |
| P A$^{122-403}$ | TPX2$^{25-45}$ | No signal | No signal | No signal |

**Figure 7**. The N-terminal half of TPX2$^{1-25}$ is the minimal region needed for binding to Aurora A. (**A**) Isothermal titration calorimetry (ITC) measurements conducted with various TPX2 constructs show that TPX2 binds with similar affinity to either the phosphorylated (P) or the dephosphorylated (deP) Aurora A and that the minimal length required for binding encompasses the first 25 residues of TPX2. (**B**) At the functional level, TPX2$^{1-25}$ can protect Aurora A from λ protein phosphatase (λPP)-directed dephosphorylation to the same extent as TPX2$^{1-45}$.
*Figure 7. Continued on next page*

*Figure 7. Continued*

The following figure supplements are available for figure 7:

**Figure supplement 1**. The first 25 amino acids of TPX2 bind similarly to either dephosphorylated or phosphorylated Aurora A.

**Figure supplement 2**. Two conserved tyrosines of TPX2 nestled inside a hydrophobic pocket in Aurora A trigger allosteric activation.

in *Dodson et al. (2013)* were started by serial dilutions from a high enzyme stock solution. The ongoing discussions on a large collection of functional, structural, and computational data on autophosphorylation in protein kinases underscore both the evolution of differential regulation mechanisms and the difficulty of elucidating these complex biological mechanisms.

Our findings may have more general implications for the family of eukaryotic Ser/Thr kinases. A number of swapped dimer structures have been solved from this enzyme class, and while *Dodson et al. (2013)* argued that these dimers are crystallographic artifacts (such as Chk2), our correlation between a domain-swapped dimer and its functional relevance would instead suggest that the reported structures are mechanistically meaningful. For direct evidence the functional relevance of these dimer structures would need to be assessed, as has been reported by elegant experiments for Chk2 (*Oliver et al., 2006*; *Pike et al., 2008*). Interestingly, most of the known swapped dimers sit in an inactive conformation (using all structural hallmarks for an active kinase discussed in *Figure 2*) (*Figure 3–figure supplement 1*). However, a few dimers capture all active-state structural signatures (*Figure 3–figure supplement 1*, LOK and Chk2 [*Oliver et al., 2006*; *Pike et al., 2008*]). All previously reported Aurora A dimers are in inactive conformations (PDB IDs 2BMC [*Fancelli et al., 2005*], 3DJ5, and 3DJ6), while in our new structure both Aurora A molecules are in the active conformation with one monomer serving as the enzyme molecule and the other as the substrate.

The functional effect of autophosphorylation is a 100-fold increase in the catalytic activity of Aurora A. While this catalytic boost is comparable to other Ser/Thr kinases (*Hagopian et al., 2001*; *Prowse and Lew, 2001*; *Adams, 2003*), the more surprising result has been our finding that TPX2 binding activates dephosphorylated Aurora A to similar levels. This is again in contrast to previous reports of additive effects from the two distinct activation mechanisms (*Dodson and Bayliss, 2012*). Our results point to a classic allosteric regulation mechanism where either phosphorylation in the activation loop or TPX2 binding in the conserved remote hydrophobic groove shifts the equilibrium far towards the active state.

While the work here has only been concerned with in vitro experiments, it may provide insight into the regulatory roles of Aurora A in the cellular context. Immunofluorescence data have shown that the centrosome-associated Aurora A pool is mainly phosphorylated, whereas the spindle-associated and TPX2-bound Aurora A is dephosphorylated (*Tyler et al., 2007*; *Scutt et al., 2009*; *Sloane et al., 2010*; *Zeng et al., 2010*). In parallel, a recent study on the *Caenorhabditis elegans* homologue of Aurora A kinase, AIR-1, showed that spindle-microtubule associated Aurora A was not phosphorylated and could nonetheless carry on centrosome-independent microtubule formation (*Toya et al., 2011*). In light of our new findings that TPX2 fully activates dephosphorylated Aurora A, the previous in vivo experiments can be re-interpreted as kinase-activity dependent functions of spindle-microtubule associated Aurora A, and not a kinase-independent function (*Toya et al., 2011*).

Our in vitro data together with previous in vivo results suggest that nature has evolved two distinct regulation mechanisms for Aurora A in different locations within the cell: autophosphorylation as activation in the centrosomes to promote phosphorylation of downstream targets, and TPX2-mediated activation at the spindle microtubules promoting Aurora A activity to another subset of downstream targets. This hypothesis is supported by the differential timing of Aurora A and TPX2 availability. Aurora A kinase levels are available as early as S-phase and peak in the G2 phase (*Dutertre et al., 2002*; *Liu and Ruderman, 2006*). On the other hand, TPX2 levels peak in the prometaphase/spindle formation stage that follows the G2 phase (*Gruss et al., 2002*; *Eckerdt and Maller, 2008*; *Macurek et al., 2008*). Our findings help shed light on an elegant strategy for fine-tuning cellular kinetics that provides more complex regulation in higher organisms.

We finally want to raise the question whether Aurora A autophosphorylation is physiologically relevant in cells or whether phosphorylation-mediated activation is primarily accomplished by upstream

kinases. While we do not have a conclusive answer to this challenging question, which has been tackled for a number of other protein kinases, we discuss several conjectures. Although the upstream PAK1 kinase can phosphorylate Aurora A at T288, autophosphorylation appears to be the essential mode of activation because PAK1 inhibition does not abolish cell division but Aurora A inhibitors do (*Zhao et al., 2005*; *Liu and Ruderman, 2006*; *Kelly et al., 2011*). Are the in vitro autophosphorylation kinetics reported here compatible with the known phosphorylation kinetics of Aurora A during the cell cycle? Our data imply that the initial rate of Aurora A autophosphorylation is very slow because this reaction occurs via the long-lived swapped dimer between two dephosphoryated Aurora A molecules. Further autophosphorylation displays a strong sigmoidal character revealing much faster kinetics of Aurora A phosphorylation by an already phosphorylated Aurora A molecule (*Figures 5 and 6*). In light of the increased local concentration of Aurora A in the centrosomes (*Glover et al., 1995*) (although the exact concentration is not known), our measured autophosphorylation kinetics is qualitatively in line with the progression of Aurora A T288 phosphorylation during the 3–4 h of G2/M duration in HeLa cells at 37°C (*Crosio et al., 2002*; *Littlepage and Ruderman, 2002*; *Ohashi et al., 2006*).

## Materials and methods

### Cloning and purification of Aurora A kinase

TEV-cleavable, His6-tagged Aurora A kinase, either long (122–403) or short (125–392) constructs, were cloned into pET28a and expressed in Rosetta 2 (DE3) *E. coli* cells (Stratagene) for 13–15 h at 21°C. Cells were centrifuged at 5000 rpm for 15 min, resuspended in buffer A, and sonicated in the presence of EDTA-free protease inhibitor cocktail and DNAse for 4 min (20 s on, 20 s off, 3.0 V). Lysates thus obtained were filtered using a 0.22 μm filtering unit and passed through a NiNTA column. The protein was eluted at 20% buffer B and Aurora A kinase fractions were pooled and TEV-cleaved overnight at 4°C in a 5 kDa dialysis cassette that was exchanged against buffer C. Cleaved Aurora A was passed through another nickel column to remove any uncleaved reactants and His6-TEV-protease, and then purified to homogeneity through a 26/60 S200 size exclusion column. Protein thus produced was aliquoted and flash-frozen before being stored at −80°C and used for kinase assays. Mutant Aurora A$^{122-403}$ T288V was also purified the same way. The phosphorylation of all Aurora A samples including mutant forms used here were quantitatively confirmed by mass spectrometry (MS).

Dephosphorylated Aurora A kinase was obtained through a λPP co-expression system. Codon-optimized Aurora A$^{122-403}$ in pET28a and untagged λPP in T7-7 plasmid were co-transformed in BL21(DE3) cells and spread on Kan/Amp 2× YT plates. The most robust colony was used for a 2× YT pre-culture and later on to inoculate a 1 L culture to an OD of 0.2. Cells were induced with 0.6 mM IPTG for 5 h at 37°C. It was noticed that although Aurora A could grow reasonably well in LB media, λPP could not; hence, the choice of 2× YT media for all co-expression needs. Purification involved the NiNTA column, followed by overnight TEV cleavage and GST-λPP treatment, in tandem NiNTA-GST columns and finally a 26/60 S200 size exclusion column. MS was used to confirm that Aurora A kinase was completely dephosphorylated. At the end of the purification, Aurora A was dialyzed against buffer C, flash-frozen with liquid nitrogen into 1 mL aliquots and stored at −80°C.

The buffers used were:

Buffer A: 50 mM TrisHCl (pH 8.0), 300 mM NaCl, 40 mM imidazole, 20 mM MgCl$_2$, 10% (vol/vol) glycerol
Buffer B: 50 mM TrisHCl (pH 8.0), 300 mM NaCl, 500 mM imidazole, 20 mM MgCl$_2$, 10% (vol/vol) glycerol
Buffer C: 20 mM TrisHCl (pH 7.0), 200 mM NaCl, 20 mM MgCl$_2$, 5 mM TCEP, 10% (vol/vol) glycerol
Buffer D: 135 mM NaCl, 3 mM KCl, 8 mM Na$_2$HPO$_4$, 1.5 mM KH$_2$PO$_4$, 5 mM TCEP, 10% (vol/vol) glycerol, pH 7.40
Buffer E: 135 mM NaCl, 3 mM KCl, 8 mM Na$_2$HPO$_4$, 1.5 mM KH$_2$PO$_4$, 5 mM TCEP, 10% (vol/vol) glycerol, 10 mM glutathione, pH 7.40

Typical yields were 8–10 mg of phosphorylated Aurora A and 45–50 mg of dephosphorylated Aurora A (expressed in the presence of λPP) per liter of *E. coli* culture.

## Mass spectrometry

The LCMS system consisted of an Agilent 1200 series HPLC connected to an Agilent series 6520 ESI Q-TOF. Protein samples (10 µM) dissolved in a 5% acetonitrile–0.1% formic acid buffer were separated on a C18 Poroshell 300SB column (1 mm × 75 mm × 5 µm) at 0.5 mL min$^{-1}$ using a linear gradient of 5–70% acetonitrile in 0.1% formic acid. MS data were collected up to 3000 m/z and raw spectra were deconvoluted using the maximum entropy algorithm of Agilent Masshunter version B.03.01 software. External mass calibration was performed using a mixture of purine (121 m/z) and HP-0921 (922 m/z) immediately prior to measuring protein samples.

## In vitro kinase assays

Aurora A, either phosphorylated/dephosphorylated wild type or mutant protein, was mixed with either AP (APSSRRTTLCGTL), Kemptide (LRRASLG), or Lats2 (ATLARRDSLQKPGLE), in the absence or presence of 50 µM TPX2 in kinase buffer (20 mM TrisHCl, 200 mM NaCl, 3% [vol/vol] glycerol, 20 mM MgCl$_2$, 1 mM TCEP, pH 7.50). These substrates comprise the consensus sequence for Aurora A ([R/K/N]-R-X-[S/T]-B where B is any hydrophobic residue with the exception of Pro) (*Ferrari et al., 2005*; *Ohashi et al., 2006*; *Sardon et al., 2010*). Peptides were ordered through Genscript. The reaction was initiated with the addition of 5 mM ATP. Then 5 µl time points were collected, resuspended in 10 µl 6% (vol/vol) trichloroacetic acid (in water) to quench the reaction, and neutralized with 50 µl 100 mM KH$_2$PO$_4$, pH 8.0 to provide the appropriate pH for nucleotide separation. The mixture was then passed through a 0.22 µm SpinX column to remove any protein precipitation. Reverse phase-high performance liquid chromatography (RP-HPLC) and an ACE 5 C18-AR, 100 Å pore size column, were used to separate nucleotides as well as peptides. For nucleotide runs, 2 µl of the aforementioned mixture was sufficient for analysis, whereas for the peptide runs the optimal injection volume was 20 µl. Nucleotide runs were routinely performed to ensure no unproductive hydrolysis was occurring during the experiment. An isocratic elution run in 100 mM KH$_2$PO$_4$, pH 6.0 was performed for this purpose. For the peptide runs, a gradient of 0–30% of elution buffer lasting 10 min at 0.4 mL/min was sufficient to separate phosphorylated from non-phosphorylated species. The running buffer was 0.1% TFA (vol/vol) in water, while the elution buffer was 100% acetonitrile. Representative peptide RP-HPLC traces are shown in *Figure 1–figure supplement 5*. Lastly, to ensure full saturation of Aurora A by TPX2 and test these proteins were well behaved, a dose-dependence curve of the effect of TPX2 on Aurora A as shown in *Figure 1–figure supplement 6* was obtained.

## Isothermal titration calorimetry

All titrations were carried out using Nano ITC (TA Instruments) and analyzed via the NanoAnalyze software using the independent fit model. Injectant was added in 1 µl volume, every 180 s, with a constant stirring speed at 350 rpm and at 25°C. Prior to ITC titration, both protein and peptide were dialyzed/resuspended in 20 mM TrisHCl, 200 mM NaCl, 3% (vol/vol) glycerol, 1 mM TCEP, pH 7.50. The concentrations used for each of the runs in *Figure 7A* were: 35 µM dephosphorylated (deP) A$^{122–403}$ + 280 µM TPX2$^{1–147}$, 48 µM deP A$^{122–403}$ + 680 µM TPX2$^{1–45}$, 13 µM deP A$^{122–403}$ + 250 µM TPX2$^{1–25}$, 18 µM deP A$^{122–403}$ + 250 µM TPX2$^{25–45}$, 20 µM phosphorylated (P) A$^{122–403}$ + 280 µM TPX2$^{1–147}$, 90 µM P A$^{122–403}$ + 940 µM TPX2$^{1–45}$, 20 µM P A$^{122–403}$ + 300 µM TPX2$^{1–25}$, and 18 µM P A$^{122–403}$ + 300 µM TPX2$^{25–45}$.

## Crystallographic methods

Crystals of dephosphorylated Aurora A$^{122–403}$ in complex with AMPPCP and TPX2$^{1–45}$ were grown at 18°C by vapor diffusion and the hanging drop method. A 2:1 ratio of protein mixture:mother liquor was obtained by combining 300 µM (10 mg/ml) deP Aurora A$^{122–403}$ + 1.5 mM AMPPCP + 300 µM TPX2$^{1–45}$ with 0.2 M lithium sulfate monohydrate, 0.1 M BisTris, pH 5.5, 25% PEG3350. Similarly, crystals of dephosphorylated Aurora A$^{122–403}$ in complex with AMPPCP were obtained by mixing a 2:1 ratio of 570 µM (18 mg/ml) deP Aurora A$^{122–403}$ + 1 mM AMPPCP with mother liquor (0.2 M ammonium sulfate, 0.2 M TrisHCl, pH 7.50, 30% (wt/vol) PEG3350). These latter crystals were also grown at 18°C by vapor diffusion and the hanging drop method. The protein, peptide, and nucleotide were originally stored in 20 mM TrisHCl, 200 mM NaCl, 10% (vol/vol) glycerol, 20 mM MgCl$_2$, 1 mM TCEP, pH 7.50.

Diffraction data were collected at 100 K at Advanced Light Source (Lawrence Berkeley National Laboratory) beamlines (8.2.1 and 8.2.2). The details of data collections are listed in *Table 1*. Data were processed with the automated data reduction program X$_{IA}$2 (*Winter, 2010*) that is part of the CCP4 suite (*Winn et al., 2011*) and uses iMOSFLM (*Battye et al., 2011*) for integration and Scala (*Evans, 2006*) for scaling. Initial phases were obtained by molecular replacement (CCP4 program MOLREP [*Vagin, 1997*]) by using an Aurora A kinase structure (PDB ID 1MQ4) as a search model. The refinement was carried

out with REFMAC5 (*Murshudov et al., 2011*) and PHENIX.REFINE (*Adams et al., 2010*), followed by manual rebuilding in COOT (*Emsley and Cowtan, 2004*; *Emsley et al., 2010*).

## Analytical ultracentrifugation

Sedimentation velocity runs were performed on a Beckman Optima XL-A Analytical Ultracentrifuge at 50,000 rpm and 18°C (same as crystallization temperature). Sedimentation of 100 µM deP Aurora $A^{122-403}$ (or 150 µM deP Aurora $A^{122-403}$, or 160 µM Aurora $A^{125-392}$ D274A, or 100 µM Aurora $A^{125-392}$ C290A) + 500 µM AMPPCP and/or 100 µM $TPX2^{1-25}$ was followed at three different wavelengths (285 nm, 290 nm, and 295 nm). Data were analyzed using the SEDFIT software (*Schuck, 2000*; *Dam and Schuck, 2004*) and the continuous size-distribution option.

## Small-angle X-ray scattering (SAXS)

All SAXS experiments were done on a BioSAXS-1000 system at Brown University, Providence, RI, USA (camera length 480.3 mm, Pilatus 100 K detector). SAXS data were recorded for Aurora A at concentrations between 0.33 and 6.6 mg/ml at 20°C with 1 mg/mL $TPX2^{1-25}$ each. The momentum transfer axis (s = 4πsinθ/λ, where 2θ represents the scattering vector s and λ = 1.54187 nm) was calibrated by using silver behenate as standard. The experiment time was between 15 min and 6 h per sample, depending on the protein concentration. Data reduction of the raw image files and conversion into scattering curves was done with the SAXSLab software (Rigaku). The SAXS curves were further processed (buffer subtraction, correction for unbound TPX2) with the program PRIMUS (*Konarev et al., 2003*). We used calculated SAXS curves (program CRYSOL [*Svergun, 1995*]) from the X-ray structures of this study as reference for the monomeric and dimeric state. The amount of dimers was calculated by using a script based on the least squares method calculations.

## Targeted molecular dynamics simulations

The crystal structure of dephosphorylated Aurora A bound to AMPPCP and TPX2 (PDB ID 4C3P) was used as the starting point for building the model presented in *Figure 3B*. The electron density for the amino acids in the region 283–288 of monomer II was not distinguishable from noise. We used the tools in the software package Modeller 9.11 (*Eswar et al., 2006*) to model the missing residues. The lowest energy model was then used as the starting point for a molecular dynamics simulation run, in which the distance between the oxygen in the sidechain hydroxyl group of T288 and the γ-phosphate of the AMPPCP moiety bound to monomer I was reduced to 3 Å. To achieve this, the structure was parameterized with the CHARMM 22-protein all-atom force field with the CMAP backbone energy correction (*MacKerell et al., 1998*, *2001*). The system was solvated in a rectangular box with TIP3 water molecules and neutralized with NaCl counterions. The final simulation box contained approximately 65,000 atoms. Periodic boundary conditions were applied to the simulation box.

After energy minimization, the simulation box was gradually heated to 300 K with a time step of 1 fs while gradually reducing positional restraints in an MD simulation of 2 ns with the software NAMD 2.8 (*Phillips et al., 2005*). The system was then equilibrated for 10 ns in the NPT ensemble (T = 300 K, p = 1.01325 bar) with the software NAMD, using the Langevin dynamics method for controlling temperature, and the combined Langevin piston Nose–Hoover method for equilibrating pressure (*Martyna et al., 1994*; *Feller et al., 1995*). We then used the software GROMACS 4.5.5 (*Hess et al., 2008*) with the steered molecular dynamics functionality as implemented in the extension PLUMED 1.3 (*Bonomi et al., 2009*) to progressively reduce the distance between the hydroxyl group of T288 and the γ-phosphate of the AMPPCP moiety bound to monomer I. This distance was reduced to 3 Å within a simulation run of 2 ns.

## Acknowledgements

We would like to thank A Gronenborn for many helpful discussions at the beginning of this project and for her gracious support in the later stages of this work. We thank N Reiter and A Rosenzweig for kindly providing the λPP plasmid, J Gelles for help with the analytical ultracentrifugation runs, and D Theobald for suggestions on the X-ray data analysis. We thank M Clarkson for the access to the BioSAXS-1000 beamline at Brown University, Providence, RI, USA and the Advanced Light Source, Berkeley, CA, USA for access to beamlines BL8.2.1. and BL8.2.2. The Berkeley Center for Structural Biology is supported in part by the National Institutes of Health, National Institute of General Medical Sciences, and the Howard Hughes Medical Institute. The Advanced Light Source is supported by the Director, Office of Science, Office of Basic Energy Sciences, of the U.S. Department of Energy under Contract No. DE-AC02-05CH11231. The simulations were performed using the resources provided via the Teragrid account TG-MCB090166T.

## Additional information

### Funding

| Funder | Grant reference number | Author |
|---|---|---|
| Howard Hughes Medical Institute | | Dorothee Kern |
| National Institutes of Health | GM100966-01 | Dorothee Kern |
| U.S. Department of Energy | The Office of Basic Energy Sciences, Catalyst Science Program, DE-FG02-05ER15699 | Dorothee Kern |

The funder had no role in study design, data collection and interpretation, or the decision to submit the work for publication.

### Author contributions

AZ, VB, Conception and design, Acquisition of data, Analysis and interpretation of data, Drafting or revising the article; SK, NK, FP, Conception and design, Acquisition of data, Analysis and interpretation of data; Y-JC, Acquisition of data, Analysis and interpretation of data; DK, Conception and design, Analysis and interpretation of data, Drafting or revising the article

## Additional files

### Major datasets

The following datasets were generated:

| Author(s) | Year | Dataset title | Dataset ID and/or URL | Database, license, and accessibility information |
|---|---|---|---|---|
| Zorba A, Kutter S, Kern D | 2014 | Structure of dephosphorylated Aurora A (122-403) bound to TPX2 and AMPPCP | http://www.pdb.org/pdb/search/structidSearch.do?structureId=4c3p | Publicly available at RCSB Protein Data Bank. |
| Zorba A, Kutter S, Cho Y-J, Kern D | 2014 | Structure of dephosphorylated Aurora A (122-403) bound to AMPPCP | http://www.pdb.org/pdb/search/structidSearch.do?structureId=4c3r | Publicly available at RCSB Protein Data Bank. |

The following previously published datasets were used:

| Author(s) | Year | Dataset title | Dataset ID and/or URL | Database, license, and accessibility information |
|---|---|---|---|---|
| Cheetham GMT, Knegtel RMA, Coll JT, Renwick SB, Swenson L, Weber P, Lippke JA, Austen DA | 2002 | Crystal structure of Aurora-2, an oncogenic serine/threonine kinase | http://www.pdb.org/pdb/explore/explore.do?structureId=1MUO | Publicly available at RCSB Protein Data Bank. |
| Heron NM, Anderson M, Blowers DP, Breed J, Eden JM, Green S, Hill GB, Johnson T, Jung FH, Mcmiken HHJ, Mortlock AA, Pannifer AD, Pauptit RA, Pink J, Roberts NJ, Rowsell S | 2006 | Aurora A kinase activated mutant (T287d) in complex with ADPNP | http://www.pdb.org/pdb/explore/explore.do?structureId=2c6d | Publicly available at RCSB Protein Data Bank. |
| Fancelli D, Berta D, Bindi S, Cameron A, Cappella P, Carpinelli P, Catana C, Forte B, Giordano P, Giorgini ML, Mantegani S, Marsiglio A, Meroni M, Moll J, Pittala V, Roletto F, Severino D, Soncini C, Storici P, Tonani R, Varasi M, Vulpetti A, Vianello P | 2005 | Aurora-2 T287d T288d complexed with PHA-680632 | http://www.pdb.org/pdb/explore/explore.do?structureId=2bmc | Publicly available at RCSB Protein Data Bank. |

| | | | | |
|---|---|---|---|---|
| Nowakowski J, Cronin CN, McRee DE, Knuth MW, Nelson C, Pavletich NP, Rodgers J, Sang BC, Scheibe DN, Swanson RV, Thompson DA | 2002 | Crystal structure of Aurora-A protein kinase | http://www.pdb.org/pdb/explore/explore.do?structureId=1mq4 | Publicly available at RCSB Protein Data Bank. |
| Bayliss R, Sardon T, Vernos I, Conti E | 2003 | Structure of human Aurora-A 122-403 phosphorylated on thr287, thr288 | http://www.pdb.org/pdb/explore/explore.do?structureId=1ol7 | Publicly available at RCSB Protein Data Bank. |
| Bayliss R, Sardon T, Vernos I, Conti E | 2003 | Structure of Aurora-A 122-403, phosphorylated on thr287, thr288 and bound to TPX2 1-43 | http://www.pdb.org/pdb/explore/explore.do?structureId=1ol5 | Publicly available at RCSB Protein Data Bank. |
| Pike ACW, Rellos P, Niesen FH, Turnbull A, Oliver AW, Parker SA, Turk BE, Pearl LH, Knapp S | 2008 | Crystal structure of human ste20-like kinase (diphosphorylated form) bound to 5- amino-3-((4-(aminosulfonyl)phenyl)amino)-n-(2,6- difluorophenyl)-1h-1,2,4-triazole-1-carbothioamide | http://www.pdb.org/pdb/explore/explore.do?structureId=2jfl | Publicly available at RCSB Protein Data Bank. |
| Pike ACW, Rellos P, Niesen FH, Turnbull A, Oliver AW, Parker SA, Turk BE, Pearl LH, Knapp S | 2008 | Crystal structure of human ste20-like kinase bound to 5-amino-3-((4-(aminosulfonyl) phenyl)amino)-n-(2,6-difluorophenyl)-1h-1,2,4-triazole-1-carbothioamide | http://www.pdb.org/pdb/explore/explore.do?structureId=2j51 | Publicly available at RCSB Protein Data Bank. |
| Oliver AW, Paul A, Boxall KJ, Barrie SE, Aherne GW, Garrett MD, Mittnacht S, Pearl LH | 2006 | Crystal structure of human Ck2 in complex with ADP | http://www.pdb.org/pdb/explore/explore.do?structureId=2cn5 | Publicly available at RCSB Protein Data Bank. |
| Pike ACW, Rellos P, Niesen FH, Turnbull A, Oliver AW, Parker SA, Turk BE, Pearl LH, Knapp S | 2008 | Crystal structure of human serine threonine kinase-10 bound to su11274 | http://www.pdb.org/pdb/explore/explore.do?structureId=2j7t | Publicly available at RCSB Protein Data Bank. |
| Pike ACW, Rellos P, Niesen FH, Turnbull A, Oliver AW, Parker SA, Turk BE, Pearl LH, Knapp S | 2008 | Crystal structure of human zip kinase in complex with a tetracyclic pyridone inhibitor (pyridone 6) | http://www.pdb.org/pdb/explore/explore.do?structureId=2j90 | Publicly available at RCSB Protein Data Bank. |
| Sunami T, Byrne N, Diehl RE, Funabashi K, Hall DL, Ikuta M, Patel SB, Shipman JM, Smith RF, Takahashi I, Zugay-Murphy J, Iwasawa Y, Lumb KJ, Munshi SK, Sharma S | 2010 | Crystal structure of unphosphorylated p70S6K1 (Form I) | http://www.pdb.org/pdb/explore/explore.do?structureId=3a60 | Publicly available at RCSB Protein Data Bank. |
| Lee SJ, Cobb MH, Goldsmith EJ | 2008 | Crystal structure of domain-swapped OSR1 kinase domain | http://www.pdb.org/pdb/explore/explore.do?structureId=3dak | Publicly available at RCSB Protein Data Bank. |

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
