## [Decision Letter]

Thank you for sending your work entitled “Molecular mechanism of Aurora A kinase autophosphorylation and its allosteric activation by TPX2” for consideration at *eLife*. Your article has been favorably evaluated by a Senior editor and 3 reviewers, one of whom, Philip Cole is a member of our Board of Reviewing Editors.

The Reviewing editor and the other reviewers discussed their comments before we reached this decision, and the Reviewing editor has assembled the following comments to help you prepare a revised submission.

The manuscript by Zorba et al is a multifaceted study aimed at clarifying the roles of autophosphorylation and TPX2 mechanisms of activation of the Aurora A kinase (AAK) as well as the molecularity of autophosphorylation of AAK. Several key findings include:

1) The stimulation by AAK autophosphorylation of Thr288 (by 100-fold) is roughly matched by TPX2 stimulation of unphosphorylated AAK (by 40-fold)

2) The domain swapped dimer of the unphosphorylated AAK in the presence of TPX2 represents a possible interaction of intermolecular autohosphorylation.

3) The solution phase assays especially with a dead mutant as substrate demonstrate the kinetic competence of intermolecular autophosphorylation.

Additional bolstering data include molecular modeling showing how the disordered loop in monomer II of AAK could be positioned for phosphoryl transfer in the active site of monomer I of AAK, analytical ultracentrifugation and SAXS showing the presence of a minor population of solution phase dimer in AAK, and ITC and enzymatic experiments demonstrating that aa1-25 of TPX2 are necessary and sufficient for high affinity binding (ca 2 uM) and activation. The careful use of Abs and mass spec really help to provide quantitative information about the state of Thr288 phosphorylation in the various enzyme forms. Importantly, the results here fill gaps in the literature about AAK regulation and contradict those from a recent study from Dodson et al reporting that AAK autophosphorylation is intramolecular. Zorba et al point out several limitations in Dodson et al that can account for the differing conclusions. Overall, Zorba et al is carefully executed, well-written, and makes a set of significant findings that should interest a broad readership. We ask the authors to address the issues below.

1) A missing piece that if added would significantly enhance this study relates to the structural basis of AAK autophosphorylation. The domain swapped AAK dimer shows surfaces of interaction in the C-lobes that one assumes are important for intermolecular autophosphorylation if the authors' model is correct. It would be very useful to mutagenize these contacting residues to ascertain their contributions to the kinetics of AAK autophosphorylation. A hypothesis to be tested would be that once autophosphorylation is completed (even if slower in such contact mutants), activity of such AAK mutants with peptide substrates would be near normal.

2) The data in Figure 5 at 1 µM AAK suggests a half-life of at least 400 seconds. This would predict a half-life of at least 4000 seconds (67 minutes) at 100 nM. These slow rates invite concerns about the physiologic significance of autophosphorylation as an activation mechanism of AAK rather than its activation by an upstream kinase such as PAK1 or CDK11. On the other hand, it appears that the assays were carried out at 25°C, which could slow things down. Moreover, the local effective concentration of AAK may be higher than its total cellular concentration. We hope that the authors can provide a perspective on whether the preponderance of existing evidence supports that AAK autophosphorylation is relevant to AAK's normal cellular function.

---

## [Author Response]

*1) A missing piece that if added would significantly enhance this study relates to the structural basis of AAK autophosphorylation. The domain swapped AAK dimer shows surfaces of interaction in the C-lobes that one assumes are important for intermolecular autophosphorylation if the authors' model is correct. It would be very useful to mutagenize these contacting residues to ascertain their contributions to the kinetics of AAK autophosphorylation. A hypothesis to be tested would be that once autophosphorylation is completed (even if slower in such contact mutants), activity of such AAK mutants with peptide substrates would be near normal*.

This is a very good suggestion that would provide compelling additional evidence for the functional role of the swapped dimer. We had originally thought about this experiment but were discouraged by the fact that most of the intermolecular interactions for the dimer are present as corresponding intramolecular contacts in the monomer and by our pervious failure to disrupt a dimer interface in another kinase. Challenged by the reviewers, we gave this experiment a try and to our pleasant surprise it worked quite cleanly with the C290A mutation. Residue C290 makes different contacts in the swapped dimer relative to the monomer, and we therefore speculated that a mutation in this position might not be too disruptive to the monomer fold and activity.

The C209A mutant is predominantly monomeric (by AUC) and indeed severely hampered in its autophosphorylation. As logically reasoned by the referees, the activity of the phosphorylated C290A mutant towards peptide substrates is comparable to the WT protein. We added these new data and the interpretation to the manuscript (in the Results and Discussion section of the manuscript and Figure 6).

*2) The data in*
Figure 5
*at 1 µM AAK suggests a half-life of at least 400 seconds. This would predict a half-life of at least 4000 seconds (67 minutes) at 100 nM. These slow rates invite concerns about the physiologic significance of autophosphorylation as an activation mechanism of AAK rather than its activation by an upstream kinase such as PAK1 or CDK11. On the other hand, it appears that the assays were carried out at 25°C*, *which could slow things down. Moreover, the local effective concentration of AAK may be higher than its total cellular concentration. We hope that the authors can provide a perspective on whether the preponderance of existing evidence supports that AAK autophosphorylation is relevant to AAK's normal cellular function*.

This is an excellent point! Although we do not have a definite answer to this challenging but crucial question, we have added one paragraph to the end of the Conclusion rationalizing that autophosphorylation could conceivably be a major mechanism in vivo.

A) Pak1 inhibition only delays the cell cycle, but does not arrest it and does not lead to cell death, in contrast to specific Aurora A kinase inhibitors that do.

B) As rationalized by the referees, the increased local concentration of Aurora A and the increased cellular temperature conjectures that our in vitro-measured autophosphorylation kinetics at 25°C are compatible with known Aurora A T288 phosphorylation kinetics during the 3 to 4 hours of G2/M duration in HeLa cells.

Furthermore we discuss the strong sigmoidal phosphorylation kinetics measured in our manuscript as additional evidence.